# A deep learning system for heart failure mortality prediction

**Dengao Li**[1,2]*, **Jian Fu**[1,2], **Jumin Zhao**[3], **Junnan Qin**[4], **Lihui Zhang**[5]

**1** College of Data Science, Taiyuan University of Technology, Taiyuan, China, **2** Technology Research Center of Spatial Information Network Engineering of Shanxi, Jinzhong, China, **3** College of Information and Computer, Taiyuan University of Technology, Taiyuan, China, **4** Department of Cardiology, Shanxi Academy of Medical Sciences, Tongji Medical College, Shanxi Bethune Hospital, Shanxi Medical University, Tongji Shanxi Hospital, Huazhong University of Science and Technology, Taiyuan, China, **5** Department of General Medical, Shanxi Academy of Medical Sciences, Tongji Medical College, Shanxi Bethune Hospital, Shanxi Medical University, Tongji Shanxi Hospital, Huazhong University of Science and Technology, Taiyuan, China

* lidengao@tyut.edu.cn

**Data Availability Statement:** Our data is not available upon request from the MIT Laboratory for Computational Physiology Institutional Data Access / Ethics Committee. Our data is from the MIMIC-III (Medical Information Mart for Intensive Care III). We obtain permission to use MIMIC-III

## Abstract

Heart failure (HF) is the final stage of the various heart diseases developing. The mortality rates of prognosis HF patients are highly variable, ranging from 5% to 75%. Evaluating the all-cause mortality of HF patients is an important means to avoid death and positively affect the health of patients. But in fact, machine learning models are difficult to gain good results on missing values, high dimensions, and imbalances HF data. Therefore, a deep learning system is proposed. In this system, we propose an indicator vector to indicate whether the value is true or be padded, which fast solves the missing values and helps expand data dimensions. Then, we use a convolutional neural network with different kernel sizes to obtain the features information. And a multi-head self-attention mechanism is applied to gain whole channel information, which is essential for the system to improve performance. Besides, the focal loss function is introduced to deal with the imbalanced problem better. The experimental data of the system are from the public database MIMIC-III, containing valid data for 10311 patients. The proposed system effectively and fast predicts four death types: death within 30 days, death within 180 days, death within 365 days and death after 365 days. Our study uses Deep SHAP to interpret the deep learning model and obtains the top 15 characteristics. These characteristics further confirm the effectiveness and rationality of the system and help provide a better medical service.

## Introduction

Heart failure (HF) is a condition that causes structural or functional abnormalities of the heart through a variety of causes, resulting in dysfunction of the ventricular systolic or diastolic functions [1]. It is the final development stage of various heart diseases [2]. According to the American College of Cardiology, cardiovascular disease causes one-third of the world's death. More than five million people in the United States suffer from heart failure, and 550,000 new cases are diagnosed each year [3–5]. Meanwhile, in China the prevalence of HF for people over

according to the requirements and downloaded the corresponding data. All who want to use the data must be a credentialed user and sign the data use agreement for the project from https://physionet.org/ website. We provide data extraction code linked https://github.com/foneone/SQL-code-for-extracting-heart-failure-from-MIMIC which can be used by the authorized user to extract related data.

**Funding:** The article is supported by the following projects: National Major Scientific Research Instrument Development Project (grant number 62027819): High-speed Real-time Analyzer for Laser Chip's Optical Catastrophic Damage Process, awarded to JZ; The General Object of National Natural Science Foundation(grant number 62076177): Study on the risk Assessment Model of heart failure by integrating multi-modal big data, awarded to DL; Shanxi Province key technology and generic technology R&D project (grant number 2020XXX007): Energy Internet Integrated Intelligent Data Management and Decision Support Platform, DL; Key research and development program of Shanxi Province (grant number 202102020101006), awarded to JZ.

**Competing interests:** The authors have declared that no competing interests exist.

35 years old is 1.3%, and there are approximately 8.9 million HF patients [6]. As a part of cardiovascular disease, HF is an important cause of rising global mortality and has become a major public health problem worldwide [7]. Because of the high prevalence, the unsatisfactory prognosis and high re-hospitalization rate of HF, the direct and indirect heart failure costs are estimated to be \$29 billion per year [8]. Therefore, an effective mortality prediction can help the doctor make more scientific treatment plans and prevent it from worsening, so as to improve the quality of life and reduce the medical expenses.

In recent years, there have been mortality prediction systems [9, 10] based on machine learning and deep learning algorithm, which could be widely used and make better analysis of the medical information. For example, Mark Stampehl et al. [11] applied three machine learning methods: classification and regression trees (CART), full logistic regression, and stepwise logistic regression, for giving the hospitalized Medicare patients a good mortality prediction. The ROC is greater than or equal to 0.74. Similarly, Sho Suzuki et al. [12] used a multiple stepwise logistic regression to select factors that influence HF mortality. Furthermore many improved tree models were used to predict mortality, such as CART [13], Random Forest (RF) [14, 15], Gradient Boosted Classification Tree (GBM) [16, 17]. In addition Zhe Wang et al. [18] utilized multiple empirical kernel learning to measure the weight of each feature influenced mortality. Besides compared with machine learning, deep learning can automatically select the important features and gain a better performance. Joon-myoung Kwon et al. [19] established a deep neural network (DNN) to predict mortality in patients with acute heart failure. Compared with the Guidelines–Heart Failure (GWTG-HF) score and other models, the deep learning model has a better AUC(0.880). Because convolutional neural network (CNN) has demonstrated excellent performance in handling features Zhe Wang et al. [20] used CNN to deal with high-dimensional features.

However, the above model ignores that in practice the collected dataset has the problems of incompleteness and imbalance. And it seriously influences the model accuracy and performance. Therefore, a **D**eep **L**earning **S**ystem based on **M**ulti-head **S**elf-attention **M**echanism (DLS-MSM) is proposed to better solve those problems. In the system data processing stage, we introduce an indicator vector to mark whether the value is a padding value, so as to quickly process the missing and expand the dimension. Then we use a convolutional network with different kernel sizes to represent features. But, after the recombination of convolution kernels of different sizes, the feature dimensions expand, which makes it difficult to determine which features should be paid more attention to. Hence we introduce the multi-head attention mechanism on CNN to better deal with characteristics from multiple views. Besides, the Focal loss function is introduced to pay more attention to imbalanced small samples and indistinguishable sample data, which solves the system data imbalance problem. Based on MIMIC-III public database data and Deep SHAP theory, the experiment proves that proposed system can effectively predict the four types of HF dead patients, who dead within 30 days, 180 days, 365 days and dead after 365 days.

## Materials and methods

### The system framework

The framework adopted in this paper shows in Fig 1. The system has four parts, which are (1) Data extraction, (2) Data preprocessing, (3) Deep learning model and (4) Prediction results. Frist of all, we extract the HF patient information from the MIMIC-III databases. Then, we propose to set the indicator vector for missing values flag after been filled. Finally, a CNN deep learning model based on multi-head self-attention methods (DLS-MSM) is proposed for mortality prediction.

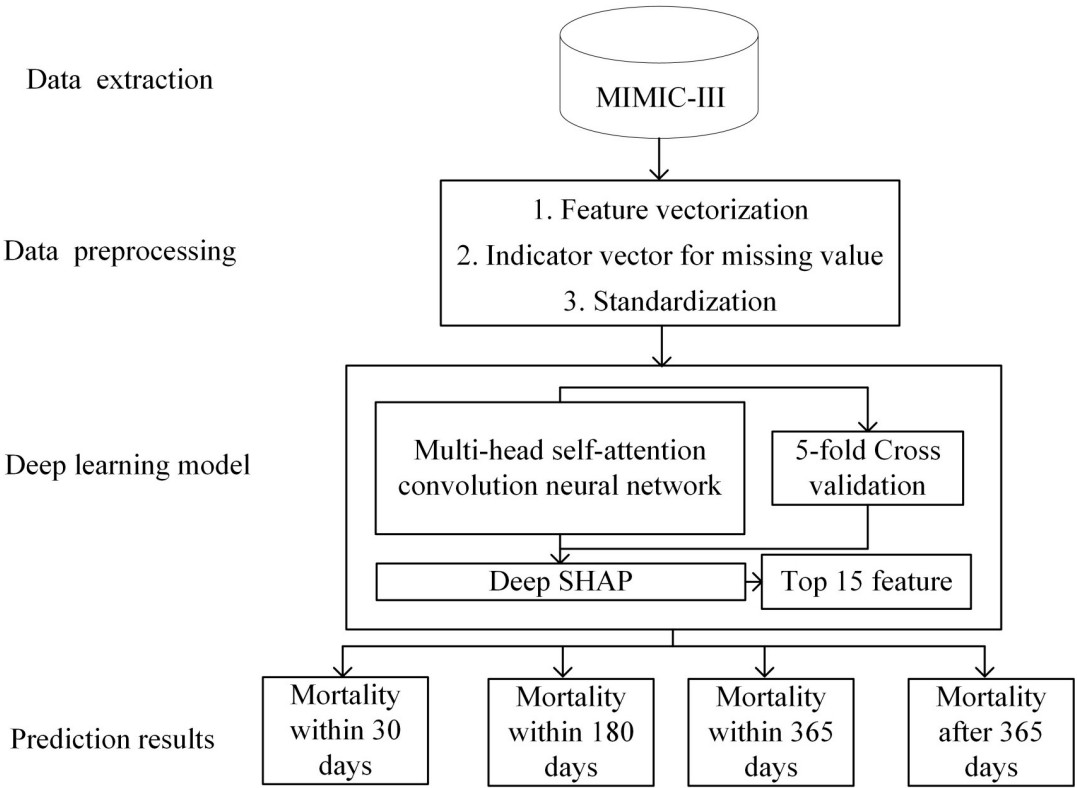

**Fig 1. The system framework, including data extraction, data preprocessing, deep learning model and prediction, results in four parts.**

## Data extraction

The HF dataset is extracted from the MIMIC-III v1.4. MIMIC-III (Medical Information Mart for Intensive Care III) [21, 22] is a large, freely-available database comprising health-related data associated with over forty thousand patients who stayed in critical care units of the Beth Israel Deaconess Medical Center between 2001 and 2012. MIMIC-III V1.4 adopted ICD-9 codes. According to ICD-9 codes, 25 types of heart failure were extracted, shown in Table 1.

Under the criterion that age greater than or equal to 18 years, we totally extracted 10311 patients. Starting point is defined as the time of first hospitalized HF patients, and the end-point is the time when patients were dead or discharge. Then, each patient has a death time, such as 0, 364. Zero means alive. The number 364 means that a patient died 364 days after being admitted to the hospital for the first time. So, we divide HF patient by death time into five categories: survivable patients who are alive in the statistical period, dead within 30 days patients, dead within 180 days patients, dead within 365 days patients and patients died after 365 days. Each group of died patients form a group with those who did not. Therefore, we have four binary experimental classifications. Every group data is a binary value {0, 1}. All patients died were labeled as a positive sample, others is negative sample, shown in Table 2.

Besides we calculate the imbalanced rate (IR) defined in the formula (1). Four datasets of imbalanced rate respectively is 1.7148, 3.0431, 7.9981 and 2.5277. The W365D dataset is the

**Table 1. HF patients correspond to the ICD-9 codes.**

| ICD-9 codes | Name |
|---|---|
| 39891 | Rheumatic heart failure (congestive) |
| 40201 | Malignant hypertensive heart disease with heart failure |
| 40211 | Benign hypertensive heart disease with heart failure |
| 40291 | Unspecified hypertensive heart disease with heart failure |
| 40401 | Hypertensive heart and chronic kidney disease, malignant, with heart failure and with chronic kidney disease stage I through stage IV, or unspecified |
| 40403 | Hypertensive heart and chronic kidney disease, malignant, with heart failure and with chronic kidney disease stage V or end stage renal disease |
| 40411 | Hypertensive heart and chronic kidney disease, benign, with heart failure and with chronic kidney disease stage I through stage IV, or unspecified |
| 40413 | Hypertensive heart and chronic kidney disease, benign, with heart failure and chronic kidney disease stage V or end stage renal disease |
| 40491 | Hypertensive heart and chronic kidney disease, unspecified, with heart failure and with chronic kidney disease stage I through stage IV, or unspecified |
| 40493 | Hypertensive heart and chronic kidney disease, unspecified, with heart failure and chronic kidney disease stage V or end stage renal disease |
| 4280 | Congestive heart failure, unspecified |
| 4281 | Left heart failure |
| 42820 | Systolic heart failure, unspecified |
| 42821 | Acute systolic heart failure |
| 42822 | Chronic systolic heart failure |
| 42823 | Acute on chronic systolic heart failure |
| 42830 | Diastolic heart failure, unspecified |
| 42831 | Acute diastolic heart failure |
| 42832 | Chronic diastolic heart failure |
| 42833 | Acute on chronic diastolic heart failure |
| 42840 | Combined systolic and diastolic heart failure, unspecified |
| 42841 | Acute combined systolic and diastolic heart failure |
| 42842 | Chronic combined systolic and diastolic heart failure |
| 42843 | Acute on chronic combined systolic and diastolic heart failure |
| 4289 | Heart failure, unspecified |

most imbalanced.

$$IR = \frac{Total\ negative\ samples}{Total\ positive\ samples} \tag{1}$$

## Data preprocessing

**Feature vectorization.** From MIMIC-III, we obtain 66 features including 39 discrete, 3 complete continuous and 24 having missing values continuous features. First we vectorize 39

**Table 2. Group and number of experiments.**

| Groups, alias | Positive samples{1} | Negative samples{0} | Total | IR |
|---|---|---|---|---|
| 1, W30D | 2472, within 30 days patients | 4239, survivable patients | 6711 | 1.7148 |
| 2, W180D | 1393, within 180 days patients | 4239, survivable patients | 5632 | 3.0431 |
| 3, W365D | 530, within 365 days patients | 4239, survivable patients | 4769 | 7.9981 |
| 4, A365D | 1677, after 365 days patients | 4239, survivable patients | 5916 | 2.5277 |

discrete features by one-hot coding. There are gender (index 1 in Table 3), medication (index 3–9), surgery (index 10–13), related diseases (index 14–39) and the feature of Stayed in CCU (index 42).

As shown in Table 3, the digit 0 is male and the digit 1 is female for gender. Then according to medicine efficacy, the medications are divided into 7 groups. Those are ACEI, ARB, beta-receptor blockers, CCB, digitalis, diuretic, and nitrates. Next, the surgery contains 4 classes, which are left ventricular assistant device (LVAD), cardiac resynchronization therapy (CRT), automatic implantable cardioverter/defibrillator check (ICD) and heart transplantation. Besides, we summary 26 diseases as the related diseases (see Appendix A in S1 File), such as Cardiac arrhythmias and Cardiomyopathy. The feature of medication, surgery, related diseases and staying in CCU are represented in the same way. The digit 1 means a patient has used one of the medication or has had one of the surgery and so on. On the contrary, the digit 0 means not.

**Proposed indicator vector for missing values method.**   Since the 24 dimensions of laboratory test (see Appendix B in S1 File), heart rate and BMI features contain missing values, it is necessary to fill the missing values. Mean with variance is the most widely used missing value imputation techniques [23]. However a lot of characteristics in the MIMIC-III HF database have very large variance. For example, white cells count in blood (WBC) feature's mean is 17.1002 K/uL and variance is 15.1180. So the mean/variance method is not suitable. Hence, we straightforwardly chose the easiest way to deal with missing values, which the normal range of the characteristics is used to fill missing values, and then the filled value is marked as the padded value. The digit 1 is a flag that the value is missing and has been filled, and the digit 0 means it's a true value. In this approach, the problem of incomplete database can be solved quickly.

A sample processed by filling method is shown in Table 4. The hemoglobin value is missing, so we use the random value in the normal range to fill. Meanwhile we set digit 1 as an indicator value, which points out hemoglobin value is filled and not true value. Heart rate in Table 4 is true value, so the indicator vector is 0. Other features have implications similar to hemoglobin and heart rate.

Since an indicator value is added after all missing features, the final feature dimension is 90 showed in Table 3. To further illustrate the features, Table 5 displays the composition of features. After missingness imputation, we adopt the Z-score normalization method to normalize data for avoiding the influence of outliers and extremes.

**Table 3.  The total features of HF patients.**

| Index | Name | Representation | Meaning and scope |
|---|---|---|---|
| 1 | Gender | One-hot coding, {0, 1} | 0 means male, 1 means female |
| 2 | Age | Value | 18–91.4 |
| 3–9 | Medication | One-hot coding, {0, 1} | 1 means used, 0 means not |
| 10–13 | Surgery | One-hot coding, {0, 1} | 1 means has had surgery, 0 means not |
| 14–39 | Related diseases | One-hot coding, {0, 1} | 1 means with this related diseases, 0 means not |
| 40 | ICU turnover times | Value | 0–41 |
| 41 | ICU stay time | Value | 0–268.25 |
| 42 | Stayed in CCU | One-hot coding, {0, 1} | 1 means stayed, 0 means not |
| 43–86 | Laboratory test | Value and one-hot coding, {0, 1} | — |
| 87–88 | Heart rate | Value and one-hot coding, {0, 1} | 30–303 |
| 89–90 | BMI | Value and one-hot coding, {0, 1} | 11.7–78.4 |

**Table 4. The variables having missing value are preprocessed.**

| WBC | Indicator value | Hemoglobin | Indicator value | . . . | Heart rate | Indicator value | BMI | Indicator value |
|---|---|---|---|---|---|---|---|---|
| 8.2 | 0 | 15.47 | 1 | . . . | 86 | 0 | 28.9609 | 0 |

## Deep learning model

**CNN with different kernel sizes.** A convolutional neural network (CNN) is a feedforward neural network. Compared with the multi-layer perceptron (MLP) [24], CNN has the local connectivity which can make use of the local information, optimize network parameters and structure, reduce model training time and improve performance [25]. Therefore, we use CNN to predict the mortality. Because, the dataset, in our research, is a piece of HF patient characteristic information, so the input is in a 1D format. Therefore, we adopt the 1D convolution. Furthermore, the 1D convolution only convolutes one direction (vertical) of features sequence. It is easy to separate the indicator vector from the indicated feature and hard to capture characteristic information. Therefore, our study uses multiple size kernels to extract key information for better capturing the information of the indicator vector and local correlation.

In conclusion, the deep learning model is indicated in Fig 2. Firstly, the input size is 256*(1*90). 256 is the batch size. And 90 is the feature dimension and the reason has been explained in the section of *Proposed indicator vector for missing values method*. Secondly, eight convolutional kernels of (1*3) size combine with Batch Normalization (BN) and ReLU function as Convolution layer. Thirdly three different convolution groups, 3*(1*3), 3*(1*4) and 3*(1*5) combine with BN, ReLU function and MaxPooling as different kernels layer, which respectively receive the convolution layer learned information. Furthermore, the MaxPooling with 2 units of sampling window is used. Fourthly, stitching layer binds every output from different kernels structure, changing to 256*396. Then the multi-head self-attention structure is used to focus on the global and local information from the stitching layer. We set the head in multi-head self-attention is two. The reason is discussed in subsection 4.2. Fifthly, the fully connected layer with (396 128) and (128 2) size processes information from the attention mechanism. Finally the output is 256*(1*2) which means we obtain 256 mortality predictions at one time.

**Multi-head self-attention mechanism.** Attention simulates the human brain processing mechanism. It identifies the target area, through focusing on crucial information and neglecting other information. In this way, the efficiency and accuracy of a model can make great progress [26]. Yingying Zhang et al. [27] proposed that representation in different subspaces likely focuses on different information. All subspaces can enhance the global information. This idea inspires us. There are three different size receptive fields in our model, and each group focuses on different information. After splicing, all information is summarized. The inside of the information is a relatively complete whole, while there is no correlation and interaction between the three cores. Hence, we use the multi-head self-attention mechanism illustrated in Fig 3 to obtain the global information and pay more attention on the important features.

As shown in Fig 3 the x (3*132) in stitching layer goes through three different linear layers to generate keys (denoted as K), queries (denoted as Q) and values (denoted as V), which is described in the formula (2). And the self-attention reflects the input from K, Q and v is the

**Table 5. The composition of features.**

| Discrete features | Continuous features | | Indicator vectors | Total features |
|---|---|---|---|---|
| | Complete features | Missing features | | |
| 39 | 3 | 24 | 24 | 90 |

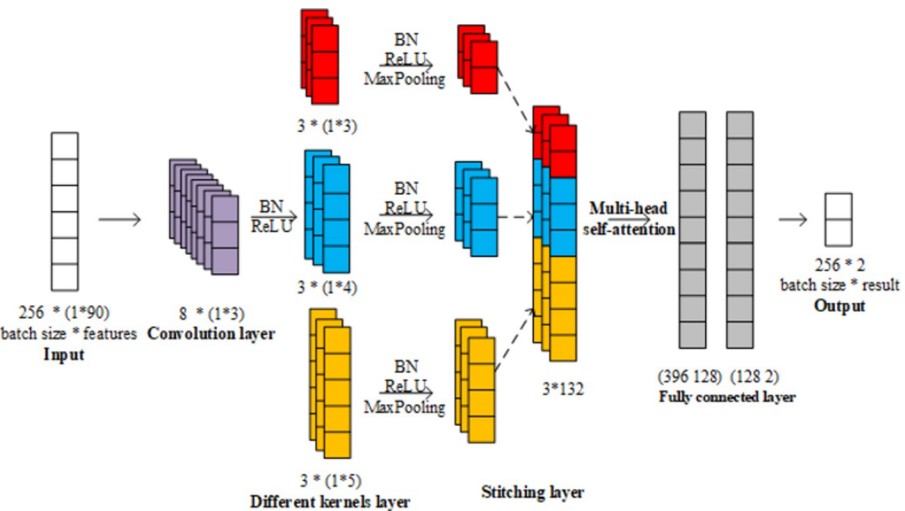

**Fig 2. The CNN based on multi-head self-attention for prediction.**

same.

$$K = W^k x + b_k$$
$$Q = W^q x + b_q \tag{2}$$
$$V = W^v x + b_v$$

Here, $(W^k, b_k)$ is a set of parameters about a linear layer also named fully connected layer. And $W^k$ is the weight and $b_k$ is the bias. The $(W^q, b_q)$ and $(W^v, b_v)$ is the same as the $(W^k, b_k)$.

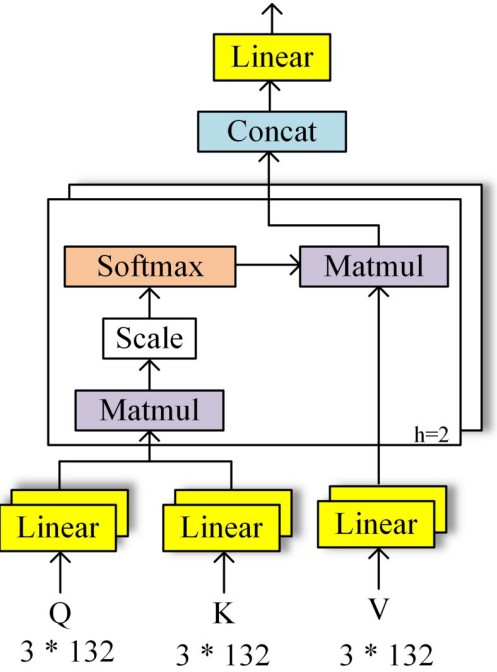

**Fig 3. The architecture of multi-head self-attention mechanism.**

Then, the attention calculates the similarity between Q and K. The similarity reflects the importance of the extracted V, that is, the weight. Then according to input dimension (denoted as d_model), it scales the weights. Next, the attention value is obtained by weighted average, using the softmax function. The self-attention is reflected in Q = K = V. Formula (3) demonstrates the same process.

$$attention(Q, K, V) = softmax(\frac{QK^T}{\sqrt{d\_model}}) \times V \tag{3}$$

Afterwards, the multi-head self-attention mechanism used different head (number of h) to gain different representations from (Q, K, V). Ultimately, it concatenates the different results through a linear layer.

$$head_i = attention(Q_i, K_i, V_i) \tag{4}$$

$$multihead = concat(head_1, \ldots, head_h)W^o \tag{5}$$

Where $head_i$ is the i-th head. In our research, h is equal to 2. And through the all steps, the output size is $3^*132$.

**Focal loss function.** In general, the imbalanced problem is a common problem in medical data processing and analysis. Similarly, the class imbalanced problem exists in MIMIC-III shown in Table 2. Therefore, we apply the focal loss function to deal with the problem. Focal loss function is proposed for one-stage detector in image object detection [28]. By reducing the weight of a large number of negative samples in the training samples, focal loss function makes the model focus on the category with fewer samples in the training process. Meanwhile, by reducing the weight of samples that are easy to classify, the accuracy of difficult to classify samples is improved.

The focal loss function is developed from the cross entropy loss function. And the cross entropy is defined:

$$CE(p, y) = \begin{cases} -log(p) & y = 1 \\ -log(1 - p) & otherwise \end{cases} \tag{6}$$

where y is {0,1} and denotes the true label in dataset. In this research, label 1 is the dead HF patient. And $p\epsilon[0,1]$ is the model prediction. For simplification, the transformation is as follow.

$$p_t = \begin{cases} p, & if\ p = 1 \\ 1 - p & otherwise \end{cases} \tag{7}$$

Hence, the cross entropy loss is defined as follows:

$$CE(p_t) = \log(p_t) \tag{8}$$

From the formula (8), the focal loss function representation is as follows.

$$FL(p_t) = -\alpha_t(1 - p_t)^\gamma \log(p_t) \tag{9}$$

In formula (8), the $\alpha_t$ and γ are two hyper-parameters. The $\alpha_t$ is used to adjust the proportion of positive and negative samples. Moreover, the γ revises the samples which are difficult to separated. In this study, we set $\alpha_t$ and γ are 0.25 and 2 respectively.

## Results

### Training strategy

In first step, we adopt the 5 fold cross-validation to train our model for avoiding the over-fitting and under-fitting. Then, we divide each kind of dataset into a training set, verification set and test set. 5% of the test sets are randomly generated from each kind of dataset. Besides, the training epoch is 120. We used the Root Mean Square prop (RMSprop) optimizer and the initial learning rate is 0.001.

In the model evaluation step, we used seven criterions. There are accuracy (ACC), Positive Prediction Value (PPV) also named Precision, Negative Prediction Value (NPV), Recall, F1 score (F1), and Area Under Positive Rate (AUC), shown in formula (10) to (14). Considering the datasets are imbalanced, the model stability is crucial. Hence, we adopted the micro-average of AUC sensitive to the small samples to reflect stability.

$$ACC = \frac{TP + TN}{TP + TN + FN + TN} \tag{10}$$

$$PPV = \frac{TP}{TP + FP} \tag{11}$$

$$NPV = \frac{TN}{TN + FN} \tag{12}$$

$$Recall = \frac{TP}{TP + FN} \tag{13}$$

$$F1 = \frac{2 * TP}{FP + FN + 2 * TP} \tag{14}$$

In the above formulas, TP is the number of true positive samples, and on the contrary, TN is the number of true negative samples. FP is the number of false-positive samples and FN is the number of false-negative samples.

### Mortality prediction for HF patients

In this subsection, we use proposed model to predict HF patients' mortality. First of all, we apply model in the W30D datasets. As shown in Fig 4A, after 80 epochs, the model is gradually stable. The loss decreases from 0.9477 to 0.0417. The ACC maintains at 84.56%, and F1 score rises from 23.01% to 78.69%. The results reveal that it is of capacity that the model distinct dead patients and living patients. Besides, the AUC in Fig 5A is 91.00% declaring the model has good stability. Moreover, micro-average AUC displays 91.00%, which denotes model has the capability to deal with imbalanced data.

Secondly, we measure the model performance on W180D datasets. The loss is constant at 0.0423, through 70 epochs in Fig 4B. Compared with W30D, the imbalanced rate of W180D is higher. Therefore, ACC downs three percent and finally keeps in 82.08%. As the imbalance rate increased, the F1 score drops to 56.33%, as does the ACC. However, the model remains stable. The AUC is 82.00% of both classes, in Fig 5B. As well as, the micro-average AUC reaches 88.00%. From this phenomenon, the model still handles the imbalance datasets well.

Afterwards when the imbalance rate continues to increase in W365D, the loss becomes more volatile. The Fig 4C loss curve appears this phenomenon. The loss disturbances between 0.022 and 0.024. Along with this comes that the AUC is 75.00% of both classes in Fig 5C,

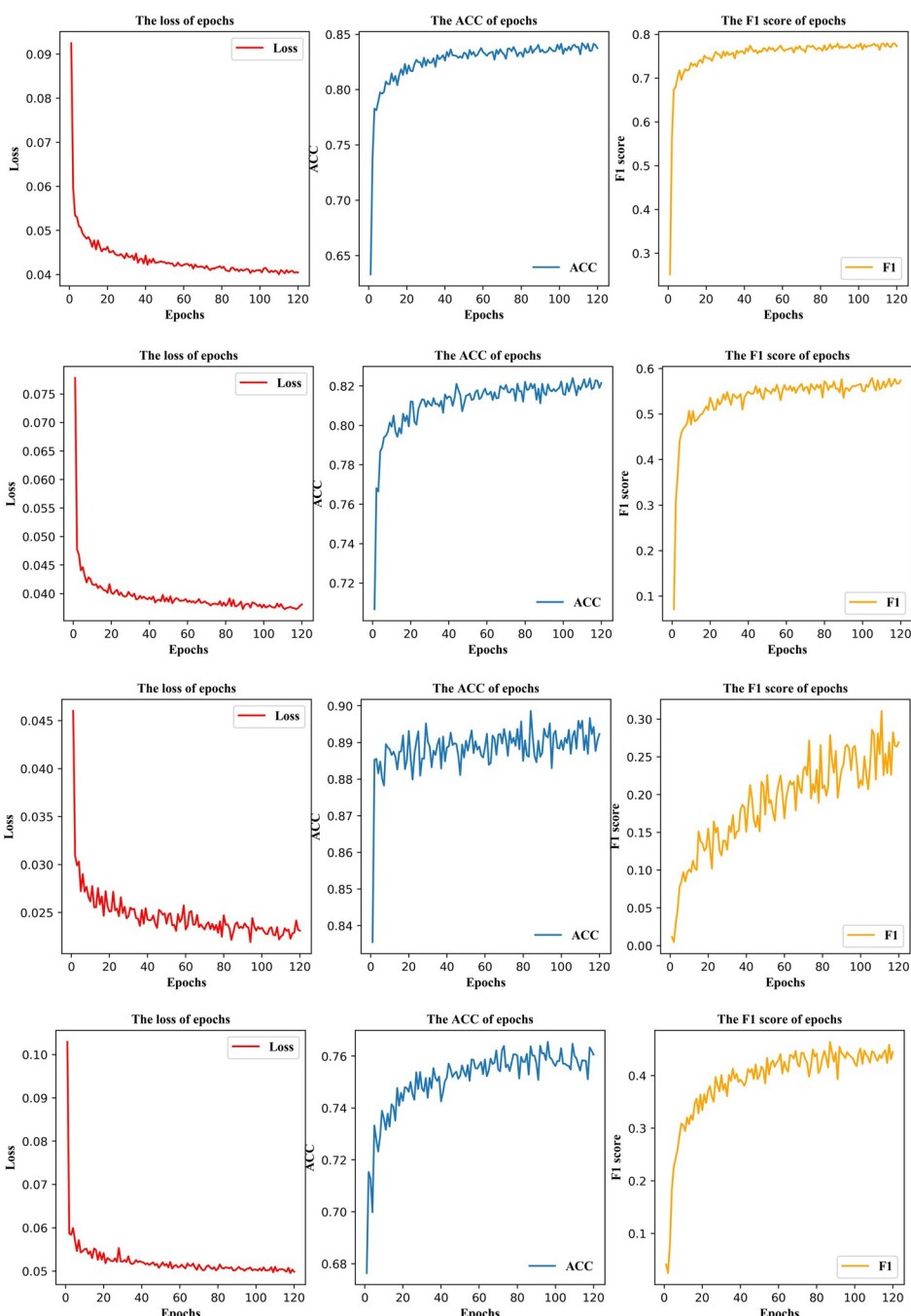

**Fig 4. The loss, ACC F1 score curve in four experimental classifications.**

which indicates the decreasing stability. Moreover, there's a huge difference between the micro-average AUC and macro-average AUC. The micro-average AUC is 91.00%. But the macro-average AUC is 75.00%. This difference indicates the model has the problem to handle the dataset with 7.988 imbalance rate. Then, although the ACC is higher than others, reaching 88.56%, the F1 score merely represented at 34.35%. This experiment proves the prediction tends to the category with a large number.

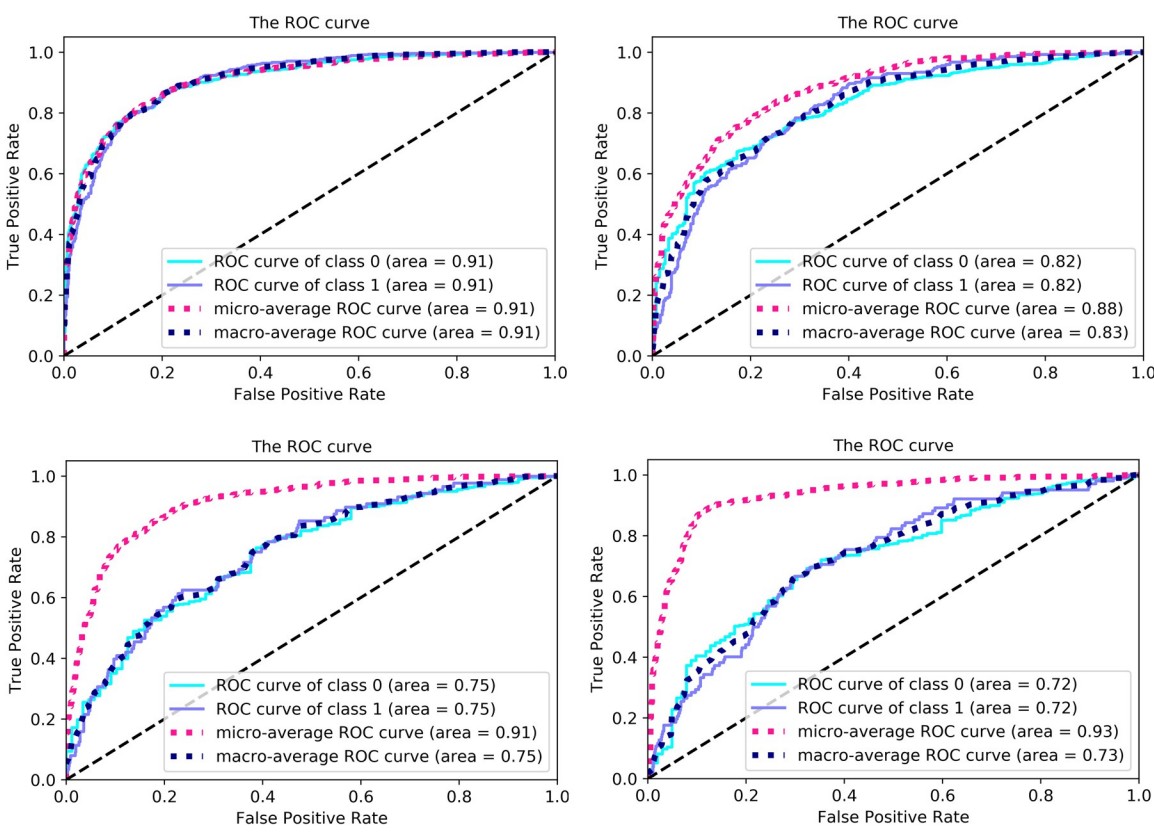

**Fig 5. The AUC in four experimental classifications.**

Ultimately, we discuss A365D dataset. The loss maintains around 0.050 in Fig 4D. It is slightly higher than others. This leads to the ACC is 76.07% and F1 is 46.34%. The A365D records HF patients who died after 1 year. However in A365D, 66.19% (1110/1677) of HF patients went 2 years or more. Therefore, there are more similarities in the characteristics of patients who died after one year and those who not died during the statistical period, comparing with other datasets. It declares the reason why the model has a lower ACC on A365D. In addition, AUC reaches of 72% both of classes. Fig 5D reports the micro-average AUC is 93%, explaining the model stability.

## Model performance compared with other methods

In order to prove the validity, we compare DLS-MSM with six comparison models, which is representative and widely applied in the medical field or bio-medical field. There are Support Vector Machine (SVM) [29], Multi-layer Perceptron (MLP) [24], Logistic Regression (LR), Random Forest (RF) [30], Light Gradient Boosting Machine (LigthGBM, LGB) [31] and K-Nearest Neighbor (KNN). Table 6 showed comparison. All the decimals have been converted into percentages. Besides, for highlighting the best results for each metric have been bolded. The effects both are from the test set.

Because W30D dataset is enough and has small imbalance ratio, the models all perform well. Specifically, RF obtains 86.31% ACC, and is 1.75% higher than DLS-MSM. In addition, the F1 score is the harmonic mean of PPV and Recall. So, F1 score make progress by 12.21%, 21.12% and 16.565%, respectively, in W180D, W365D and A365D. This phenomenon indicates that our model is preferable to deal with the dataset. As shown in Table 6, F1 score of the SVM and RF

**Table 6. Model performance compared with other methods.**

| Datasets | Measurement | DLS-MSM | SVM | MLP | LR | RF | LGB | KNN |
|---|---|---|---|---|---|---|---|---|
| **W30D** | ACC | 84.56 | 85.12 | 80.95 | 82.44 | **86.31** | 84.23 | 80.35 |
| | PPV | 76.28 | **80.64** | 70.16 | 70.96 | 72.58 | 77.40 | 64.52 |
| | NPV | 83.57 | 87.74 | 83.04 | 89.15 | **94.34** | 88.21 | 89.62 |
| | Recall | 81.26 | 79.37 | 76.32 | 79.28 | **88.24** | 79.34 | 78.43 |
| | F1 | 78.69 | **80.00** | 73.11 | 74.89 | 79.64 | 78.36 | 70.80 |
| | AUC | **91.00** | 83.97 | 79.82 | 81.64 | 86.85 | 83.15 | 79.82 |
| | Micro-average AUC | 91.00 | 90.00 | 89.00 | 90.00 | **92.00** | 91.00 | 89.00 |
| **W180D** | ACC | **82.08** | 80.14 | 76.60 | 79.79 | 79.08 | 78.72 | 80.85 |
| | PPV | **59.94** | 32.86 | 47.30 | 34.28 | 20.00 | 35.71 | 34.29 |
| | NPV | 85.21 | 95.75 | 83.32 | 94.81 | **98.58** | 92.92 | 96.23 |
| | Recall | 53.13 | 71.86 | 51.33 | 68.57 | **82.35** | 62.50 | 75.00 |
| | F1 | **56.33** | 45.10 | 49.23 | 45.71 | 32.18 | 45.45 | 47.06 |
| | AUC | **82.00** | 76.54 | 68.31 | 74.97 | 80.61 | 71.95 | 78.30 |
| | Micro-average AUC | **88.00** | 85.00 | 85.00 | **88.00** | 86.00 | **88.00** | 86.00 |
| **W365D** | ACC | 88.56 | 88.70 | 87.87 | 88.70 | 88.70 | **89.12** | 88.28 |
| | PPV | **31.03** | 0.00 | 18.52 | 3.71 | 0.00 | 3.7 | 0.00 |
| | NPV | 94.01 | 100.0 | 95.75 | **99.53** | 100.0 | 100.0 | 99.53 |
| | Recall | 38.46 | 0.00 | 41.67 | 50.00 | 0.00 | **100.0** | 0.00 |
| | F1 | **34.35** | | 25.64 | 6.90 | | 7.14 | 0.00 |
| | AUC | 75.00 | | 65.99 | 69.51 | | **93.53** | 44.32 |
| | Micro-average AUC | 91.00 | | 91.00 | **93.00** | | 92.00 | 90.00 |
| **A365D** | ACC | **76.07** | 71.96 | 70.94 | 76.01 | 72.97 | 74.66 | 72.63 |
| | PPV | **47.35** | 9.52 | 35.71 | 30.96 | 4.76 | 29.76 | 22.62 |
| | NPV | 78.14 | 96.70 | 83.08 | 93.87 | **100.0** | 92.45 | 92.45 |
| | Recall | 45.37 | 53.33 | 43.42 | 66.67 | **100.0** | 60.98 | 54.29 |
| | F1 | **46.34** | 16.16 | 39.19 | 42.28 | 9.09 | 40.00 | 31.93 |
| | AUC | 72.00 | 63.14 | 63.08 | 72.05 | **86.30** | 68.92 | 64.69 |
| | Micro-average AUC | **93.00** | 74.00 | 76.00 | 82.00 | 82.00 | 82.00 | 77.00 |

model are null in W365D dataset. This indicates those models have difficulty handling imbalanced data. However, in a large number of negative samples, DLS-MSM can accurately identify the dead HF patients. Moreover, the NPV in the four algorithms are basically flat at 99.88%, which indicates algorithms applied in W365D dataset are hard to solve the imbalance problem. In this point, the DLS-MSM with highest PPV and F1 score is much better than other algorithms.

On the one hand, from the perspective of AUC, the model remains relatively stable. Because of the higher imbalance ratio in W365D, the positive samples are virtually ignored in LGB, from higher Recall and lower F1. Hence, LGB is hard to be adopted in the mortality prediction. The same explanation applies to the RF, which is to 88.70% AUC in A365D. On the other hand, the micro-average of AUC proves that DLS-MSM is effective in dealing with imbalance problem. The micro-average of AUC promotes around 1.5%.

## The effect of indicator vector

In order to verify the validity of the indicator vector, we carry out a comparative experiment. The dataset without indicator vector is the control group. So the dimension of input in the experiment is changed to 66. And the dimension of attention layer and linear layer changes to (2, 96) and (288,128) respectively.

Table 7 displays the comparative experiment, which establishes the validity of the indicator vector. All the model metrics with indicator vector are higher than without, except W365D. Because W365D dataset is the most lopsided dataset and F1 is lower than the model with indicator vector. The training model ignores the influence of the positive samples, in W365D. Adding indicator helps model to identify more positive samples from true positive samples. Furthermore the NPV value is higher than not adding the indicator vector which indicates the model can comprehensively handle the positives and negative samples.

## The effect of using attention

Attention plays an import role in the CNN framework. Therefore, we compare CNN framework with and without attention, showing in Table 8. Because PPV value and Recall are mutually impacted. Recall in DLS-MSM with attention mechanism is higher than without attention. On the contrary, PPV value in without attention model is higher than with attention. This result involves the problem of recognition rate. The higher Recall means that the model can distinguish true positive samples. The NPV value reflects that the DLS-MSM with attention can accurately identify negative samples. In addition, the model with attention is more stable, which is displayed by the AUC and Micro-average AUC. All in all the proposed DLS-MSM system with attention is better than without attention except the recall in W30D database.

## Discussion

### Effect of attention in different locations

The attention mechanism plays different roles in different locations of the system structure. Therefore, we involve two distinct experiments to study the effective locations. According to the principle of attention mechanism, we choose the position one between convolution layer and different kernels layer and the position two between the different kernel layers and stitching layer shown in Fig 2. The sizes of the attention mechanism are respectively (2, 90), (2, 44).

As shown in Table 9, recall of W180D dataset and A365D dataset in position 1 is 5.64% and 26.87% higher than position 2 respectively. However, the PPV is not higher. Hence

**Table 7. Effects before and after adding indicator vector.**

| Datasets | Measurement | Before | After | Datasets | Measurement | Before | After |
|---|---|---|---|---|---|---|---|
| **W30D** | ACC | 81.64 | **84.56** | **W365D** | ACC | 87.52 | **88.56** |
| | PPV | **82.49** | 76.28 | | PPV | **33.33** | 31.03 |
| | NPV | 79.24 | **83.57** | | NPV | **94.37** | 94.01 |
| | Recall | 68.32 | **81.26** | | Recall | 10.78 | **38.46** |
| | F1 | 74.74 | **78.69** | | F1 | 16.30 | **34.35** |
| | AUC | 79.24 | **91.00** | | AUC | 61.46 | **75.00** |
| | Micro-average AUC | 90.00 | **91.00** | | Micro-average AUC | **92.00** | 91.00 |
| **W180D** | ACC | 80.47 | **82.08** | **A365D** | ACC | 70.37 | **76.07** |
| | PPV | 56.69 | **59.94** | | PPV | 46.92 | **47.35** |
| | NPV | 84.94 | **85.21** | | NPV | 76.62 | **78.14** |
| | Recall | 51.14 | **53.13** | | Recall | 43.49 | **45.37** |
| | F1 | 53.77 | **56.33** | | F1 | 45.14 | **46.34** |
| | AUC | 71.61 | **82.00** | | AUC | 62.76 | **73.00** |
| | Micro-average AUC | 88.00 | **88.00** | | Micro-average AUC | 77.00 | **93.00** |

**Table 8. Effect of using attention and not using.**

| Position | Measurement | W30D | W180D | W365D | A365D |
|---|---|---|---|---|---|
| Without attention | ACC | 82.51 | 80.75 | 81.79 | 74.29 |
| | PPV | 75.04 | **66.27** | **31.24** | **54.72** |
| | NPV | **86.78** | 78.10 | 93.53 | 73.73 |
| | Recall | **81.35** | 43.51 | 34.19 | 37.54 |
| | F1 | 78.07 | 52.53 | 32.65 | 44.53 |
| | AUC | 81.42 | 74.90 | 60.68 | 66.78 |
| | Micro-average AUC | 90.00 | 89.00 | 90.00 | 82.00 |
| DLS-MSM | ACC | **84.56** | **82.08** | **88.56** | **76.07** |
| | PPV | **76.28** | 59.94 | 31.03 | 47.35 |
| | NPV | 83.57 | **85.21** | **94.01** | **78.14** |
| | Recall | 81.26 | **53.13** | **38.46** | **45.37** |
| | F1 | **78.69** | **56.33** | **34.35** | **46.34** |
| | AUC | **91.00** | **82.00** | **75.00** | **73.00** |
| | Micro-average AUC | **91.00** | 88.00 | **91.00** | **93.00** |

attention in position 1 pays more attention on the positive samples. And in W365D because of the high imbalanced rate the attention mechanism does not work. The NPV in our model is average 1.966% higher than position 1 and 2. It indicates our model can effectively judges negative samples. Most results in our model are better than the other two positions. The results confirm the validity of the idea in our model that attention is applied after the splicing layer.

**Table 9. The effects of different positions of multi-head self-attention.**

| Position | Measurement | W30D | W180D | W365D | A365D |
|---|---|---|---|---|---|
| Position 1 | ACC | 80.78 | 76.35 | **89.43** | 66.28 |
| | PPV | **76.94** | 51.79 | **32.53** | 38.86 |
| | NPV | 83.04 | 81.19 | 92.49 | 76.92 |
| | Recall | 71.11 | 49.62 | 11.97 | 39.48 |
| | F1 | 73.91 | 50.68 | 17.50 | 39.17 |
| | AUC | 79.91 | 67.84 | 60.31 | 57.88 |
| | Micro-average AUC | 89.00 | 84.00 | **91.00** | 73.00 |
| Position 2 | ACC | 79.37 | 77.66 | 83.55 | 73.93 |
| | PPV | 74.18 | 55.39 | 31.83 | **63.35** |
| | NPV | 81.94 | 82.35 | 91.18 | 77.02 |
| | Recall | 70.70 | 45.04 | 23.93 | 12.30 |
| | F1 | 72.40 | 49.68 | 27.32 | 20.60 |
| | AUC | 78.27 | 69.30 | 60.47 | 68.93 |
| | Micro-average AUC | 87.00 | 85.00 | 90.00 | 81.00 |
| DLS-MSM | ACC | **84.56** | **82.08** | 88.56 | **76.07** |
| | PPV | 76.28 | **59.94** | 31.03 | 47.35 |
| | NPV | **83.57** | 85.21 | **94.01** | **78.14** |
| | Recall | **81.26** | **53.13** | **38.46** | **45.37** |
| | F1 | **78.69** | **56.33** | **34.35** | **46.34** |
| | AUC | **91.00** | **82.00** | **75.00** | **73.00** |
| | Micro-average AUC | **91.00** | **88.00** | **91.00** | **93.00** |

## Effect of different number of heads

The multi-head self-attention mechanism can have different heads, which may influence the system performance. From this point we set the head 2, 3, 6, 11 and 12 for analyzing. The specific results are explained in Fig 6. Besides, the more the number of the head increase the model complexity and influence the training time. Thus, we test the average time in different heads, shown in Fig 7. The unit is seconds.

The result in the upper left of Fig 6 is W30D. Recall fluctuates widely. When the number of the head reaches 12, the recall is 85.29. But there is no significant increase in F1 relative to the other results. The high recall rate but low F1 indicates that more real negative samples are predicted to be true. The number of six heads is similar to this. And the number of 3 heads and 11 heads behaves poorly. Recall is only 72.48 and 69.06 severally. Hence, in the W30D the number of 2 head has a better manifestation. Besides it spends less time training. Other datasets in Fig 6 give us a more obvious different model performance. Comprehensive analysis indicates that in our study the number of 2 head in attention has best consequence and less time consumption.

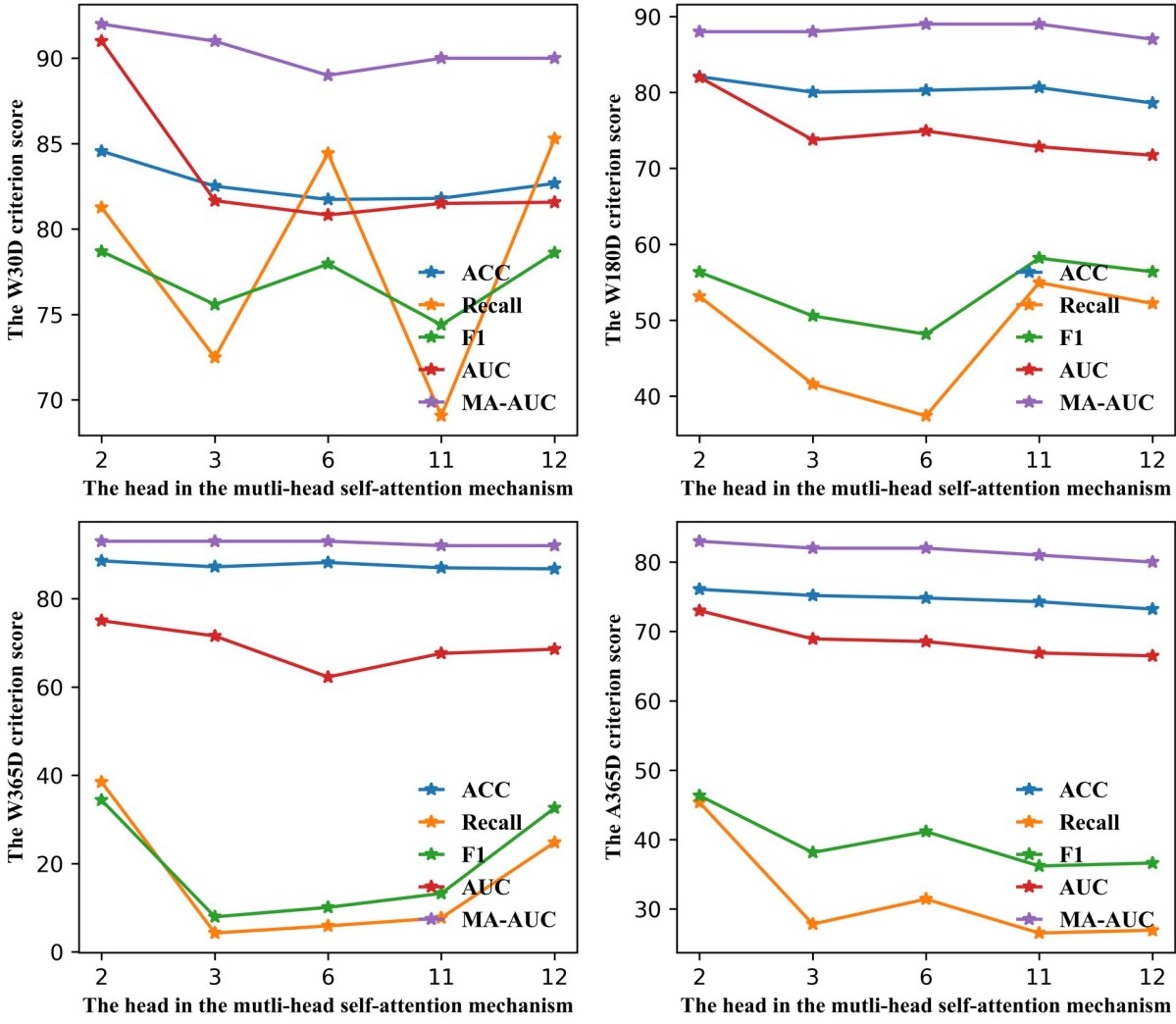

**Fig 6. Comparison of the different head in multi-head self-attention mechanism.**

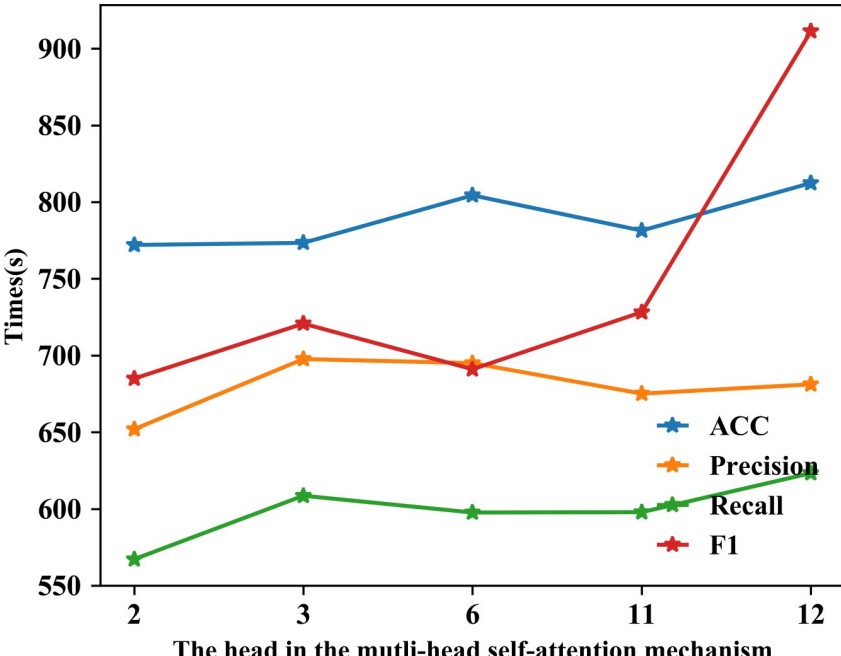

**Fig 7. Comparison of the different head results with time(s).**

## Important feature ranking based Deep SHAP

Neural network is often thought of as a black box, so how to interpret the complex models become a hot topic study. We adopt Deep SHAP [32] to interpret our deep learning model and rank the feature. Deep SHAP is a deep learning interpreted model combining DeepLIFT theory and SHAP value. DeepLIFT is used to interpret deep learning model by calculating the weight to each feature of the input in the backpropagation [33]. SHAP (SHapley Additive exPlanations) gives each feature a unique importance score. Scott M. Lundberg and Su-In Lee

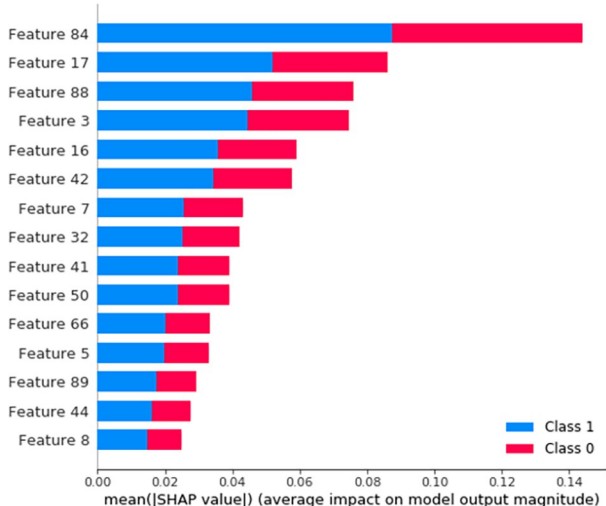

**Fig 8. The top 15 features in system.**

**Table 10. Feature correspondence q.**

| Number | Feature |
|---|---|
| Feature 84 | pCO2 (Laboratory test) |
| Feature 17 | Respiratory failure (Diagnoses) |
| Feature 88 | BMI |
| Feature 3 | ARB (Medicine) |
| Feature 16 | Pulmonary circulation disorder (Diagnoses) |
| Feature 42 | WBC (Laboratory test) |
| Feature 7 | Diuretic (Medicine) |
| Feature 32 | Diabetes (Diagnoses) |
| Feature 41 | CCU stays (ICU information) |
| Feature 50 | GLU (Laboratory test) |
| Feature 66 | Potassium (Laboratory test) |
| Feature 5 | CCB (Medicine) |
| Feature 89 | Insert BMI (Indicator vector) |
| Feature 44 | Hemoglobin (Laboratory test) |
| Feature 8 | Nitrates (Medicine) |

combine the two theories to explain the deep learning model and prove it more suitable for human to understand model. Fig 8 displays the top 15 features in our system.

In Fig 8, the blue label indicates the degree of every characteristic affects mortality. The red label indicates the alive degree. The features in Fig 8 are digital, and the characteristics corresponding to numbers are shown in Table 10.

Heart failure and severe hypoxia and ischemia can be combined with severe arrhythmia, especially the occurrence of ventricular fibrillation. Clinical symptoms include loss of consciousness, convulsions, respiratory arrest and even death. So the respiratory failure is the important features. The RAAS system is activated when heart failure occurs. ARB drugs can inhibit the RAAS system and myocardial remodeling, delay the progression of heart failure. Calcium channel blockers (CCB) can reduce calcium in myocardial cells concentration to improve myocardial active diastolic function and lower blood pressure.

From these clinical medical conclusions, it can be shown that the features extracted, in Table 10, by our system are scientific and effective. Therefore, our system can support to doctors in prognostic treatment.

## Conclusion

In our study, a CNN deep learning model based on multi-head self-attention is applied to the mortality prediction system for prognostic HF patients. The system can distinguish between four categories of death, that is, death within 30 days, death within 180 days, death within 365 days and death after 365 days. First, we proposed that the indicator vector indicates the value is true or be filled. Then a multi-head self-attention is introduced to CNN deep learning model. Finally, the Focal loss function is applied to overcome the imbalance. The results from experiment display the idea is feasible. The whole system is effective to predict the mortality. In the end in order to explain the system, we use the Deep SHAP method to make an essential feature reasonable rank.

## Supporting information

**S1 Data. SQL code.** The code is about how to extract data from the MIMIC-III.
(7Z)

**S1 File.**
(DOCX)

## Author Contributions

**Conceptualization:** Dengao Li.

**Data curation:** Jian Fu, Junnan Qin, Lihui Zhang.

**Funding acquisition:** Dengao Li.

**Investigation:** Jumin Zhao.

**Methodology:** Jian Fu.

**Project administration:** Dengao Li.

**Software:** Jian Fu.

**Validation:** Lihui Zhang.

**Visualization:** Jian Fu, Jumin Zhao, Junnan Qin.

**Writing – original draft:** Dengao Li.

**Writing – review & editing:** Jian Fu, Jumin Zhao, Junnan Qin.

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
