## [Decision Letter · Decision Letter 0]

16 Aug 2021

PONE-D-21-17467

A Deep Learning System for Heart Failure Mortality Prediction

PLOS ONE

Dear Dr. Li,

Thank you for submitting your manuscript to PLOS ONE. After careful consideration, we feel that it has merit but does not fully meet PLOS ONE’s publication criteria as it currently stands. Therefore, we invite you to submit a revised version of the manuscript that addresses the points raised during the review process.

We look forward to receiving your revised manuscript.

Kind regards,

**Le Hoang Son, Ph.D**

Academic Editor

PLOS ONE

Journal Requirements:

Whilst you may use any professional scientific editing service of your choice, PLOS has partnered with both American Journal Experts (AJE) and Editage to provide discounted services to PLOS authors. Both organizations have experience helping authors meet PLOS guidelines and can provide language editing, translation, manuscript formatting, and figure formatting to ensure your manuscript meets our submission guidelines. To take advantage of our partnership with AJE, visit the AJE website (http://aje.com/go/plos) for a 15% discount off AJE services. To take advantage of our partnership with Editage, visit the Editage website (www.editage.com) and enter referral code PLOSEDIT for a 15% discount off Editage services.  If the PLOS editorial team finds any language issues in text that either AJE or Editage has edited, the service provider will re-edit the text for free.

A clean copy of the edited manuscript (uploaded as the new *manuscript* file).

The article is supported by the following projects. (1) National Major Scientific The General Object of National Natural Science Foundation (62076177、61772358). (2) National Major Scientific Research Instrument Development Project (6202780085). (3) Shanxi Province key technology and generic technology R&D project (2020XXX007).

**Comments to the Author**

1. Is the manuscript technically sound, and do the data support the conclusions?

Reviewer #1: Partly

Reviewer #2: Partly

2. Has the statistical analysis been performed appropriately and rigorously? 

Reviewer #1: Yes

Reviewer #2: Yes

3. Have the authors made all data underlying the findings in their manuscript fully available?

Reviewer #1: Yes

Reviewer #2: No

4. Is the manuscript presented in an intelligible fashion and written in standard English?

Reviewer #1: Yes

Reviewer #2: No

5. Review Comments to the Author

**Reviewer #1**: 

In the present study, the authors presented a heart failure mortality prediction model using machine learning, which inspired many researchers. There are many models that predict the prognosis of heart failure by conventional statistical methods and currently, new predicing models using deep-learning algorithms have been introduced showing outstanding performance. In this study, the authors proposed an indicator vector to indicate whether the value is true or be padded, which fast solves the missing values and helps expand date dimensions. It appears that the study has been carefully done and the manuscript is well written and clearly presented. However, the following issues require further consideration and clarification.

First, authors should describe the development of machine learning models in detail enough for readers to reproduce the experiment.

As a major pitfall of machine learning (ML) algorithms is overfitting, external validation is needed. Without external validation, the study result is not guaranteed in other hospitals.

In order to understand the results more accurately, it is better to provide PPV, NPV of the model.

This study did not present the left ventricular function of the study population. In the study of heart failure, absence of LV ejection fraction data is a critical limitation to claim clinical value of the study. At this point, this study seems more suitable for a computer science journal than a medical journal.

**Reviewer #2**: The authors should be commended for their interesting study

However, I have some questions that I feel should be addressed

1) The introduction is too long - it could be made significantly shorter by focussing on the role of ML mortality prediction in HF

2) I was confused about how the patient population is split - it wasn't clear if subjects could be a more than 1 group - please clarify maybe with a flow diagram

3) The feature vectorisation needs more explanation - I think representative example of the feature vector (in supplemental information) would be very useful

4) There is no need to discuss CNN architecture more generally only the 1D version.

5) I was confused to how the 1x90 feature vector is created please clarify in relation to other earlier sections

6) Using a 1D CNN results in kernels crossing very different features - please discuss - why not use an MLP

7) Figure 3 seems to switch between showing inputs to kernels - maybe these should be more obviously differentiated

8) The effect of the indicator vector and self-attention head are in the discussion not the results please move to results

9) Depp SHAP is only really mentioned in the discussion - maybe more description in methods and results is required

---

## [Author Response · Author response to Decision Letter 0]

21 Dec 2021

Responses to reviewer and editor comments are too numerous to be detailed here. Please view the uploaded 'Response to Reviewers' file.

---

## [Decision Letter · Decision Letter 1]

11 Apr 2022

PONE-D-21-17467R1A Deep Learning System for Heart Failure Mortality PredictionPLOS ONE

Dear Dr. Li,

Thank you for submitting your manuscript to PLOS ONE. After careful consideration, we feel that it has merit but does not fully meet PLOS ONE’s publication criteria as it currently stands. Therefore, we invite you to submit a revised version of the manuscript that addresses the points raised during the review process.

We look forward to receiving your revised manuscript.

Kind regards,

Le Hoang Son, Ph.D

Academic Editor

PLOS ONE

**Comments to the Author**

1. If the authors have adequately addressed your comments raised in a previous round of review and you feel that this manuscript is now acceptable for publication, you may indicate that here to bypass the “Comments to the Author” section, enter your conflict of interest statement in the “Confidential to Editor” section, and submit your "Accept" recommendation.

Reviewer #2: (No Response)

Reviewer #3: All comments have been addressed

2. Is the manuscript technically sound, and do the data support the conclusions?

Reviewer #2: Partly

Reviewer #3: Yes

3. Has the statistical analysis been performed appropriately and rigorously? 

Reviewer #2: Yes

Reviewer #3: Yes

4. Have the authors made all data underlying the findings in their manuscript fully available?

Reviewer #2: Yes

Reviewer #3: Yes

5. Is the manuscript presented in an intelligible fashion and written in standard English?

Reviewer #2: Yes

Reviewer #3: Yes

6. Review Comments to the Author

**Reviewer #2**: 

I would like commend the authors the manuscript is much improved

1. However I am still concerned about the use of a CNN for the network - I agree that CNN are easier to train and it does leverage local connections - but that is not always a good thing. For instance does the order of vector features matter - what happens when convolutional kernels cross different groups. I would suggest doing a sensitivity analysis of different order of features

2. Also please put the comparison with different ML models in the results not discussion

3. The paper is still very long and needs to be edited to make it shorter

**Reviewer #3**: Publish the revised manuscript as I am satisfied with the changes made by authors. I would like to appreciate the efforts made by authors in preparing the revised version.

---

## [Author Response · Author response to Decision Letter 1]

15 Jul 2022

Thank you very much for your valuable suggestions on our paper. We have complied with journal requirements and revised these opinions one by one at the first time. More detailed modify the description can be found from the Cover Letter and Response to Reviewers.

---

## [Decision Letter · Decision Letter 2]

17 Oct 2022

A Deep Learning System for Heart Failure Mortality Prediction

PONE-D-21-17467R2

Dear Dr. Li,

We’re pleased to inform you that your manuscript has been judged scientifically suitable for publication and will be formally accepted for publication once it meets all outstanding technical requirements.

Kind regards,

**Le Hoang Son, Ph.D**

Academic Editor

PLOS ONE

Additional Editor Comments (optional):

**Comments to the Author**

1. If the authors have adequately addressed your comments raised in a previous round of review and you feel that this manuscript is now acceptable for publication, you may indicate that here to bypass the “Comments to the Author” section, enter your conflict of interest statement in the “Confidential to Editor” section, and submit your "Accept" recommendation.

Reviewer #2: All comments have been addressed

2. Is the manuscript technically sound, and do the data support the conclusions?

Reviewer #2: Yes

3. Has the statistical analysis been performed appropriately and rigorously? 

Reviewer #2: Yes

4. Have the authors made all data underlying the findings in their manuscript fully available?

Reviewer #2: Yes

5. Is the manuscript presented in an intelligible fashion and written in standard English?

Reviewer #2: Yes

6. Review Comments to the Author

**Reviewer #2**: All my comments have been addressed adequately and I have no further comments or questions about this manuscript

---

## [Editor Report · Acceptance letter]

15 Dec 2022

PONE-D-21-17467R2 

A Deep Learning System for Heart Failure Mortality Prediction 

Dear Dr. Li:

I'm pleased to inform you that your manuscript has been deemed suitable for publication in PLOS ONE. Congratulations! Your manuscript is now with our production department. 

Kind regards, 

on behalf of

Prof. Le Hoang Son 

Academic Editor

PLOS ONE